# Pre/Post-Merger Consistency Test for Gravitational Signals from Binary Neutron Star Mergers

**Matteo Breschi** [1,2,3,*] [iD], **Gregorio Carullo** [1,4] [iD] **and Sebastiano Bernuzzi** [1] [iD]

1   Theoretisch-Physikalisches Institut, Friedrich-Schiller-Universität Jena, 07743 Jena, Germany; sebastiano.bernuzzi@uni-jena.de (S.B.)
2   International School for Advanced Studies (SISSA), 34136 Trieste, Italy
3   INFN Sezione di Trieste, 34149 Trieste, Italy
4   Niels Bohr International Academy, Niels Bohr Institute, 2100 Copenhagen, Denmark
*   Correspondence: matteo.breschi@ligo.org

**Abstract:** Gravitational waves from binary neutron star (BNS) mergers can constrain nuclear models, predicting their equation of state (EOS). Matter effects on the inspiral-merger signal are encoded in the multipolar tidal polarizability parameters, whose leading order combination is sufficient to capture, with high accuracy, the key features of the merger waveform. Similar EOS-insensitive relations exist for the post-merger signal and can be used to model the emissions from the remnant. Several works suggested that the appearance of new degrees of freedom in high-density post-merger matter can be inferred by observing a violation of these EOS-insensitive relations. Here, we demonstrate a Bayesian method to test such an EOS-insensitive relation between the tidal polarizability parameters (or any other equivalent parameter) and the dominant post-merger frequency using information from the pre-and-post-merger signal. Technically, the method is similar to the inspiral-merger-ringdown consistency tests of General Relativity with binary black holes. However, differently from the latter, BNS pre/post-merger consistency tests are conceptually less informative and they only address the consistency of the assumed EOS-insensitive relation. Specifically, we discuss how such tests cannot conclusively discriminate between an EOS without respecting such a relation and the appearance of new degrees of freedom (or phase transitions) in high-density matter.

**Keywords:** gravitational waves; compact binary mergers; neutron stars; equation of state; nuclear matter





## 1. Introduction

Kilohertz gravitational waves (GWs) from binary neutron star (BNS) mergers remnants are considered a promising probe of the nuclear equation of state (EOS) at extreme densities. While no such detection was possible for GW170817 [1–3], future experiments are expected to reach the necessary sensitivity for a detection, e.g., [4–6]. Several authors claimed that a viable path to constrain the extreme-densities EOS is to "observe" specific features (e.g., frequencies) in the post-merger spectra and to employ EOS-insensitive relations (or quasi-universal relations, QUR) to unveil EOS properties (e.g., phase transitions), e.g., [7–12]. Only a few authors have, however, considered the actual observational and data analysis problem, namely, the problem of how to incorporate these speculative ideas into a rigorous Bayesian data analysis framework [8,12]. This paper discusses one possible concrete method in this direction and some related conceptual limitations in the realization of this program.

New degrees of freedom or phase transitions can impact the BNS remnant dynamics at densities $\rho \gtrsim 2\,\rho_{sat}$, where $\rho_{sat} \simeq 2.7 \times 10^{17}$ kg m$^3$ is the nuclear saturation density, and leaves signatures in the observable GWs. Case studies simulated BNSs with matter models, including hyperon production, e.g., [7,13,14] or zero-temperature models of phase transitions to quark-deconfined matter, e.g., [7,10,15,16]. In these examples, an EOS softening

with respect to the "baseline" hadronic EOS can determine a more compact remnant that either undergoes an earlier gravitational collapse or increases the post-merger GW peak frequency $f_2$ towards higher values. The former case is particularly relevant for binary masses above the prompt collapse threshold for the softened EOS, but below that threshold for the hadronic EOS. This implies that one of the two EOS models could be ruled out simply due to the observation of a post-merger signal. The latter case might instead be probed, in a suitable mass range, by observing a violation (breakdown) of the QUR that relates $f_2$ to properties of the individual neutron star (NS) in the binary, e.g., [7,8,11,17]. It is worth remarking that the detectability of these effects crucially depends on the densities at which the EOS softening takes place. Significant effects have been simulated by constructing rather "extreme" transitions.

EOS-insensitive relations are heavily used in GW astronomy with BNSs in order to either reduce the matter's degrees of freedom in waveform modeling or connect spectral features to the NS equilibria and mass-radius diagram, e.g., [18–21]. Our work focuses on the relationship between the dominant quadrupolar spectral peak of the post-merger signal, $f_2$, and the (leading order) tidal coupling constant $\kappa_2^{\mathrm{T}}$ of the binary [8,22]. This QUR allowed us to construct a unified full-spectrum model by combining an inspiral-merger tidal waveform with a post-merger completion [8,12,17,23–25]. Such a relation represents a natural (and representative) choice for a pre/post-merger (PPM) consistency test. To date, the employment of QURs is also the only method used in rigorous Bayesian studies, e.g., [6,8,12,25], to connect the binary properties to the post-merger features.

Inferring a QUR breakdown can be naturally treated as a PPM consistency test for a given QUR, similarly to analyses of binary black hole (BBH) mergers in the context of tests of General Relativity [26–28]. We naturally employ this well-established framework to the analysis of BNS transients and demonstrate how to infer a QUR breakdown using Bayesian analyses of the full BNS spectrum.

The paper is structured as follows. In Section 2, we introduce the method used to detect departures from quasi-universality. In Section 3, we validate our method performing parameter estimation (PE) on mock GW data. Finally, we provide conclusions in Section 4, highlighting conceptual issues in the interpretation of the analysis in real GW observations.

## 2. Methods

QUR breaking occurs when the quasi-universal prediction does not match the corresponding observed property. For the case of the post-merger peak $f_2(\kappa_2^{\mathrm{T}})$, the QUR is established as a function of the binary properties that can be well-estimated from pre-merger GWs. However, the post-merger signal directly provides a measurement of the $f_2$ frequency. Thus, in order to identify QUR breaks, we compare the post-merger observations to the pre-merger predictions estimated with QURs. Following the approach of Ref. [26], we introduce a consistency test that aims to reveal such breaking by employing full-spectrum observations of BNSs.

Given the GW data and a waveform template, the posterior distributions of the BNS parameters are calculated via Bayesian PE analysis (see, e.g., [29–31]). For our studies, we make use of the time-domain effective-one-body (EOB) model `TEOBResumS` [32] extended with the `NRPM` template in the high-frequency post-merger regime [8]. In order to speed up the computations, the EOB template makes use of a reduced-order approximation [33]. The considered post-merger model incorporates QURs calibrated on NR data, which are used to predict the template features, including characterization of the main peaks of the post-merger spectrum. Closely following [8], we perform three PE analyses: first, we analyze the inspiral-merger data only (labeled as 'IM') with `TEOBResumS`; then, the post-merger data only (labeled as 'PM') is studied with `NRPM`, and, finally, we perform PE on the full-spectrum data (labeled as 'IMPM') with the complete model `TEOBResumS_NRPM`.

As discussed in Ref. [26], PPM consistency tests rely on a cutoff frequency $f_{\mathrm{cut}}$ used to split the low-frequency and high-frequency regimes. In general, the time-domain post-merger signal will also include frequency contributions below the merger frequency $f_{\mathrm{mrg}}$,

due to the low quality factor of the post-merger frequencies dominating the remnant response. However, for systems dominated by the quadrupolar mode, this "mixing" is typically negligible, and the portion of the signal with $f < f_{\text{mrg}}$ only suffers from small contaminations from the time-domain post-merger phase. For this reason, choosing $f_{\text{cut}} = f_{\text{mrg}}$ is a sensible choice. The "mixing" becomes more significant for lower remnant spins (induced, e.g., by a nonspinning high mass ratio binary). We stress that, even in this case, the consistency test remains valid, although the physical interpretation of the results becomes less immediate, since a good fraction of the deviation in the $f < f_{\text{mrg}}$ region could be induced by the time-domain post-merger signal. For BNS signals, the post-merger signal can lead to significant spectral contamination below $f_{\text{cut}}$ and the split is less trivial. However, if the dominant post-merger frequencies are significantly larger than the merger frequency $f_{\text{mrg}}$ or if the post-merger signal-to-noise-ratio (SNR) contribution below the cutoff is negligible, one can still choose $f_{\text{cut}} = f_{\text{mrg}}$. This is the choice made in this work, assuming the cutoff frequency to be known exactly. In a realistic scenario, the cutoff frequency can be estimated from the full-spectrum posterior using EOS-insensitive relations for the merger frequency for the quadrupolar mode [8,24,34]. If the splitting frequency $f_{\text{cut}}$ cannot be uniquely fixed (e.g., due to spectral contamination below this threshold), the 'IM' and 'PM' models might be treated separately in single analyses either in a direct time-domain analysis [35,36], or by augmenting the standard frequency domain likelihood using "gating" techniques [36–38]. However, both of these methods are expected to significantly increase the computational cost, compounding the already long computational times inherent in inspiral BNS analyses.

The 'IM' inference provides direct information on the progenitors' properties (i.e., masses, spins, tidal polarizabilities, ...). From these parameters, it is possible to predict the $f_2$ posterior using the QUR in Equation (13) of [8]. Additionally, the 'PM' inference provides information on the progenitors' properties through the internally employed QURs. Moreover, in this case, the $f_2$ posterior can be directly estimated from the reconstructed waveform. Finally, the 'IMPM' case naturally delivers information on the progenitors' properties and it allows us to estimate the $f_2$ posterior from the reconstructed waveform. Then, following the approach of Ref. [26], we introduce the (fractional) deviations from the QUR as

$$\frac{\Delta f_2}{f_2} = \frac{f_2^{\text{PM}} - f_2^{\text{IM}}}{f_2^{\text{IMPM}}}, \quad \frac{\Delta \kappa_2^{\text{T}}}{\kappa_2^{\text{T}}} = \frac{\kappa_2^{\text{T}^{\text{PM}}} - \kappa_2^{\text{T}^{\text{IM}}}}{\kappa_2^{\text{T}^{\text{IMPM}}}}. \tag{1}$$

We remark that $f_2^{\text{IM}}$ is computed from the inspiral data using the QUR in post-processing, while $f_2^{\text{PM}}$ and $f_2^{\text{IMPM}}$ estimation includes directly the 'PM' data. On the other hand, $\kappa_2^{\text{T}}$ is directly inferred from the analyzed data for both pre-merger and post-merger studies, since it can be directly computed from the intrinsic parameter of the BNS template models.

The computation of $\text{p}(\Delta f_2 / f_2, \Delta \kappa_2^{\text{T}} / \kappa_2^{\text{T}})$ is performed with a probabilistic approach. Given the posteriors $\{f_2, \kappa_2^{\text{T}}\}_i$ for $i = \text{IM}, \text{PM}, \text{IMPM}$, the joint posterior of $\Delta f_2$ and $\Delta \kappa_2^{\text{T}}$ is estimated from the analyzed data $\boldsymbol{d}$ as

$$\text{p}(\Delta f_2, \Delta \kappa_2^{\text{T}} | \boldsymbol{d}) = \iint \text{p}(f_2, \kappa_2^{\text{T}} | \boldsymbol{d}_{\text{PM}}) \, \text{p}(\kappa_2^{\text{T}} - \Delta \kappa_2^{\text{T}}, f_2 - \Delta f_2 | \boldsymbol{d}_{\text{IM}}) \, \text{d}f_2 \, \text{d}\kappa_2^{\text{T}}, \tag{2}$$

where $\boldsymbol{d}_{\text{IM}}$ and $\boldsymbol{d}_{\text{PM}}$ are, respectively, the pre-merger and post-merger portion of data defined by the cutoff frequency $f_{\text{cut}}$. Equation (2) is the convolution product between the 'IM' and the 'PM' posteriors. Then, labeling $\varepsilon_{f_2} = \Delta f_2 / f_2$ and $\varepsilon_{\kappa_2^{\text{T}}} = \Delta \kappa_2^{\text{T}} / \kappa_2^{\text{T}}$, the posterior for the quantities in Equation (1) can be computed from the recovered posterior as

$$\text{p}(\varepsilon_{f_2}, \varepsilon_{\kappa_2^{\text{T}}} | \boldsymbol{d}) = \iint \kappa_2^{\text{T}} f_2 \, \text{p}(\varepsilon_{f_2} \cdot f_2, \varepsilon_{\kappa_2^{\text{T}}} \cdot \kappa_2^{\text{T}} | \boldsymbol{d}_{\text{IM}}, \boldsymbol{d}_{\text{PM}}) \, \text{p}(f_2, \kappa_2^{\text{T}} | \boldsymbol{d}_{\text{IMPM}}) \, \text{d}f_2 \, \text{d}\kappa_2^{\text{T}}, \tag{3}$$

where $\boldsymbol{d}_{\text{IMPM}}$ corresponds to the complete data.

As discussed in Ref. [26], $\text{p}(f_2, \kappa_2^{\text{T}} | \boldsymbol{d}_{\text{IMPM}})$ represents our best guess for the $\{f_2, \kappa_2^{\text{T}}\}$ posterior and it is used in Equation (3) to weight the contributions of the inspiral-merger

and post-merger inferences; $p(\Delta f_2, \Delta \kappa_2^T | d_{IM}, d_{PM})$ encodes the agreement/disagreement between pre-merger and post-merger inferences. Within this approach, the origin of the axes, i.e., $\Delta f_2 = 0$ and $\Delta \kappa_2^T = 0$, represents the null-hypothesis for which no deviation from quasi-universality is observed. On the other hand, a departure of the posterior from the null-hypothesis can indicate the breakdown of the $f_2(\kappa_2^T)$ QUR. Following the EOS terminology, we label as a *softening* effect a deviation towards the region with $\Delta f_2/f_2 > 0$ and $\Delta \kappa_2^T/\kappa_2^T < 0$, in order to differentiate it from a *stiffening* effect, which shows $\Delta f_2/f_2 < 0$ and $\Delta \kappa_2^T/\kappa_2^T > 0$.

## 3. Results

We demonstrate the possibility of investigating QUR breaking using PE analyses of mock GW data. We discuss the specific case of BHB$\Lambda\phi$ and DD2 EOS simulated in [14]. The BHB$\Lambda\phi$ EOS is identical to DD2 except that at densities $\rho \gtrsim 2.5\rho_{sat}$ it softens due to the formation of $\Lambda$-hyperons. Inspiral-merger GW signals from (equal-mass) binaries described by the two EOS and $M \lesssim 2.8$ M$_\odot$ are indistinguishable since the individual progenitor NSs have maximal densities $\rho \lesssim 2.5\rho_{sat}$, similar compactnesses and tidal parameters, as shown in Figure 1 (left). On the other hand, for $M \gtrsim 2.8$ M$_\odot$ the post-merger remnants reach higher densities at which the two EOS differ, leading to different post-merger GWs, as shown in Figure 1 (right). Hereafter, all reported uncertainties correspond to the 90% credibility intervals, except when explicitly mentioned.

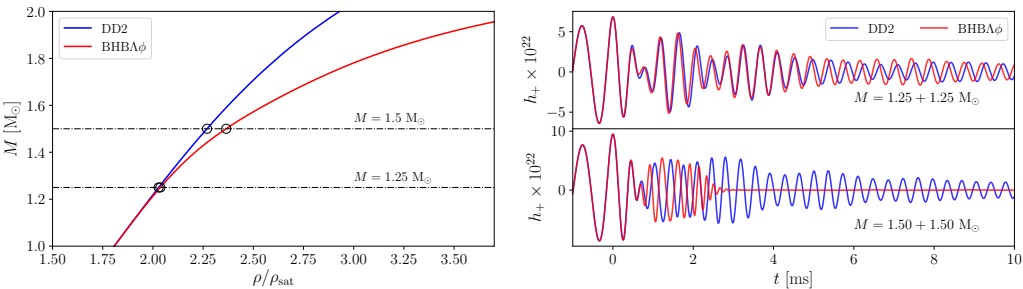

**Figure 1.** Comparison between the BHB$\Lambda\phi$ (red) and the DD2 (blue) EOS and the corresponding BNS templates [14]. (**Left panel**): Mass of individual NSs as a function of the central density. The markers refer to simulated BNSs. (**Right panel**): Plus polarization $h_+(t)$ of the NR waveforms for the simulated BNSs with mass $M = 2.5$ M$_\odot$ (top) and $M = 3$ M$_\odot$ (bottom). The binaries are located at a fiducial distance of 40 Mpc. The origin of the time axis $t = 0$ corresponds to the moment of merger.

We consider a pair of high-mass binaries with $M = 3$ M$_\odot$, no spins and equal component masses extracted from the CoRe database [39,40]. The individual progenitors of the high mass BNS have $\rho \approx 2.35\rho_{sat}$, while the associated remnant reaches $\rho \approx 2.8\rho_{sat}$ and the presence of $\Lambda$-hyperons significantly affects the post-merger dynamics. The DD2 1.50+1.50 M$_\odot$ binary has $f_2 \simeq 2.76$ kHz, and the respective BHB$\Lambda\phi$ remnant has $f_2 \simeq 3.29$ kHz (see Refs. [17,24] for discussions on the $f_2$ estimation for this case). The difference between the two NR values is ~500 Hz, which corresponds to ~20%. The BHB$\Lambda\phi$ data deviates at ~$3\sigma$ from the prediction of the QUR presented in Ref. [8] and employed in NRPM ($f_2^{fit} = 2.88$ kHz), corresponding to a more compact remnant than the DD2 case. The two binaries also have different times of black-hole collapse: the DD2 case collapses at late times, i.e., ~21 ms after merger; the BHB$\Lambda\phi$ remnant collapses shortly after merger within 2.6 ms. Moreover, we repeat the analysis on the low-mass BHB$\Lambda\phi$ binary with $M = 2.5$ M$_\odot$, whose morphology is almost identical to the corresponding DD2 case even in the post-merger phase. The corresponding waveforms are shown in Figure 1 (right). The data are generated as EOB-NR hybrid waveforms injected in zero-noise, while the recovery is performed using TEOBResumS_NRPM.

We analyze 128 s of data with a lower frequency $f_{low} = 20$Hz (or $f_{low} = f_{mrg}$ in the post-merger only case) and a sampling rate of 8192 Hz, injecting the signal with post-merger

SNR 11 (total SNR $\sim$200) and using the three-detector LIGO-Virgo network at design sensitivity [41,42]. The priors on the parameters are taken consistently with Refs. [29,30] with spin parameters fixed to zero. The PE studies are performed with the nested sampling routines implemented in `LALInference` [29,30,43] (The analysis settings are identical to Ref. [8]. There, the reader can also find detailed discussions on the posteriors)

Figure 2 shows the posterior estimated for the three considered binaries. The grey band indicates the uncertainty of the QUR, and $\Delta f_2/f_2$ posteriors falling in this band are considered to be consistent with the assumed QUR (Alternatively, at first order approximation, the QUR error can be taken into account in the $\Delta f_2$ posterior through error propagation) The low-mass BHB$\Lambda\phi$ case confidently includes the null hypothesis within the 90% credibility level of the posterior. The $\Delta f_2/f_2$ posterior for the high-mass DD2 case is fully consistent with the QUR uncertainties, indicating no significant deviation. The deviation of $\Delta\kappa_2^{\mathrm{T}}/\kappa_2^{\mathrm{T}} = 0.5^{+0.3}_{-0.3}$ toward the stiffness portion of the plane is due to the finite faithfulness of `NRPM` against the full NR simulation considered, and is expected to be cured by improved models [17,24]. The salient point to be extracted from the figure is that the high-mass BHB$\Lambda\phi$ case shows significant deviations toward the softness portion of the plane, with $\Delta\kappa_2^{\mathrm{T}}/\kappa_2^{\mathrm{T}} = -0.2^{+0.5}_{-0.2}$ and $\Delta f_2/f_2 = 0.2^{+0.2}_{-0.1}$. This deviation in the frequency is significantly above the fit uncertainty and demonstrates a successful detection of the QUR breaking, which invalidates the applicability of the QUR $f_2(\kappa_2^{\mathrm{T}})$ to the considered binary.

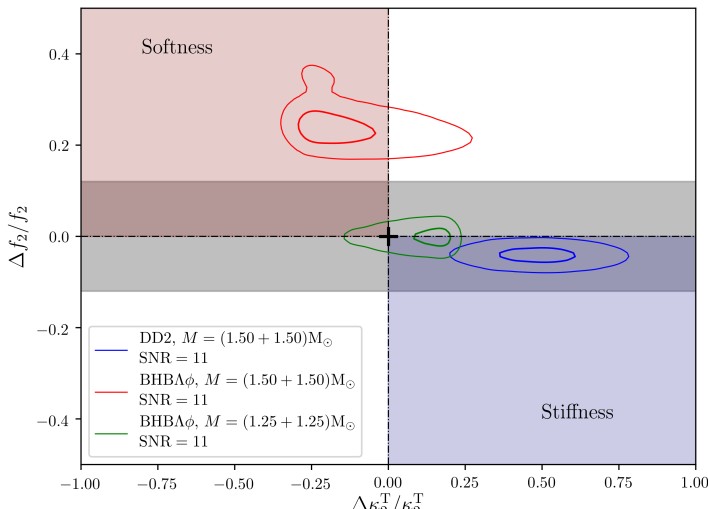

**Figure 2.** Posterior for the deviation from the quasiuniversality defined in Equation (1) for characteristic post-merger frequency $f_2$ and tidal coupling $\kappa_2^{\mathrm{T}}$. The contours report the 50% and the 90% credibility regions. Green lines refer to low-mass BHB$\Lambda\phi$ binary, blue lines refer to high-mass DD2 binary and red lines refer to high-mass BHB$\Lambda\phi$ binary. The red area denotes deviations due to softening effects, while the blue area identifies the stiffening effects. The grey band reports the 90% credible interval of the $f_2$ EOS-insensitive relation.

## 4. Conclusions

Our results demonstrate a quantitative Bayesian method to invalidate a given QUR using full-spectrum BNS observations. The observation of an inconsistency in a PPM analysis of this type might help to *exclude* (some of) the EOS employed for the design of the QUR. Although, in the specific case considered, this inconsistency was indeed caused by the appearance of hyperons at high densities (and it can be observed in other phase transitions [7,12]), we stress that demonstrating the breakdown of a QUR within a given credibility level does not necessarily imply the measurement of an EOS softening effect. Since the true EOS is not known, but the inference requires a model (the QUR) designed using an EOS sample, it is only possible to invalidate the model (hypothesis) using the proposed null test. For example, this consistency test might simply exclude a QUR which

is "not sufficiently" EOS-insensitive or is poorly designed. Refs. [11,24] discuss the specific case of $f_2(R_{1.4})$, where $R_{1.4}$ is the radius of an equilibrium NS of mass $1.4 M_\odot$. According to current available data and EOS models, the $f_2(R_{1.4})$ QUR might be easily broken by an observation at minimal post-merger SNR for detection. However, if one considers a similar QUR with the same quantities but rescaled by the binary mass, the QUR significantly improves its EOS-insensitive character. We stress that, according to current theoretical models and constraints, the breaking of a (well-designed) QUR can occur in neighboring regions of the mass parameter space, i.e., $M \gtrsim 3\,\mathrm{M_\odot}$, and for finely-tuned configurations of the equation of state. Cf. [7,10,12,14,16].

The presented method is not restricted to the particular QUR considered here. A similar analysis may be performed, for example, on the inferred collapse time [17], considering the consistency of multiple parameters/QUR involved in the GW template, or using other QURs, e.g., [7,11]. However, the $f_2(\kappa_2^{\mathrm{T}})$ QUR is particularly interesting because (i) it is directly involved in the construction of the GW template, and (ii) it is rather accurate and shows deviations at a few percent although being built from the largest sample of EOS and simulations explored so far in numerical relativity. Improved analyses can be achieved by folding-in recalibration parameters to better account for the uncertainties of the QUR, as shown in Refs. [6,17,24].

BNS post-merger signals are likely to be accessible with next-generation ground-based GW interferometers for events comparable (or louder) than GW170817, e.g., [17,44,45]. In order to gain information on the nuclear matter from these observations, it seems necessary to significantly extend current theoretical EOS models and simulations and explore such predictions within Bayesian analysis frameworks.

**Author Contributions:** Conceptualization, M.B.; Data curation, M.B.; Formal analysis, M.B.; Funding acquisition, S.B.; Investigation, M.B.; Methodology, M.B. and G.C.; Project administration, M.B.; Resources, S.B.; Validation, M.B.; Visualization, M.B.; Writing—original draft, M.B., S.B.; Writing—review & editing, G.C. All authors have read and agreed to the published version of the manuscript.

**Funding:** M.B. and S.B. acknowledge support from the European Union's H2020 ERC Starting Grant, no. BinGraSp-714626. M.B. acknowledges support by the Deutsche Forschungsgemeinschaft (DFG) under Grant no. 406116891 within the Research Training Group (RTG) 2522/1 and by the PRO3 program "DS4Astro" of the Italian Ministry for Universities and Research. G.C. acknowledges support by the Della Riccia Foundation under an Early Career Scientist Fellowship. G.C. acknowledges funding from the European Union's Horizon 2020 research and innovation program under the Marie Sklodowska-Curie grant agreement No. 847523 'INTERACTIONS', from the Villum Investigator program supported by VILLUM FONDEN (grant no. 37766) and the DNRF Chair, by the Danish Research Foundation. S.B. acknowledges support from the DFG project MEMI no. BE 6301/2-1.

**Data Availability Statement:** The waveform model employed in this work, `TEOBResumS_NRPM`, is implemented in BAJES and the software is publicly available at: https://github.com/matteobreschi/bajes . The posterior samples presented in this work will be shared on request to the corresponding author.

**Acknowledgments:** The computational experiments were performed on the TULLIO sever at INFN Turin (Italy).

**Conflicts of Interest:** The authors declare no conflict of interest.

## Abbreviations

The following abbreviations are used in this manuscript:

| | |
|---|---|
| BBH | Binary black hole |
| BNS | Binary neutron star |
| EOB | Effective-one-body |
| EOS | Equation of state |
| GW | Gravitational wave |

NR     Numerical relativity
PE      Parameter estimation
PPM   Pre/post-merger
QUR   Quasi-universal relation
SNR    Signa-to-noise ratio

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
