# Peer review of "Pre/Post-Merger Consistency Test for Gravitational Signals from Binary Neutron Star Mergers"

_2571-712X, doi:10.3390/particles6030045_

Round 1

Reviewer 1 Report

Referee report on “Pre/post-merger consistency test for gravitational signals from binary neutron star mergers’ by M. Breschi et al. submitted to Particles

The paper proposes a practical Bayesian test for the EOS-insensitive relations between the parameters of the GW signal from the binary neutron star coalescence. The method closely follows those described in Ref.[26] for the GR tests based on the black hole mergers.

The idea of the paper is straightforward and scientifically sound and the results of such a study would be definitely interesting and worth publication. However, two concerns preclude publishing of the manuscript in the present form.

First, there are some problems with numbers. The uncertainties quoted in the text do not agree with those shown in Fig. 1.

Indeed, in the final paragraph of the Results section, the authors give estimate for \delta\kappa in high-mass DD2 case as \Delta\kappa/kappa =0/5\pm 0.3. What is the credibility level of this estimate? According to the contours in Fig.2 (blue lines), the quoted uncertainties are consistent with the projection of the 2D 90% credibility region onto the \Delta\kappa axis. In the normal (Gaussian) case, this would correspond to more than 2\sigma (95.4%) 1D uncertainty. In this case, the deviation towards the stiffness region is more than 5\sigma, which is clearly not “mild”.

The same is true for the high-mass BHB\Lambda\phi numbers. The quoted \Delta f_2/f_2=0.2(+0.2,-0.1) clearly gives the 1D interval wider than the 90% credibility contour projection on the respective axis and cannot correspond to something like 1\sigma (68%) which is implicitly assumed when the uncertainties are quoted without stated credibility level (or confidence in the frequentists case).  

If the numbers in the text are correct, and correspond roughly to 1\sigma, it is not clear why the 0.5\pm 0.3 deviation from the null (~91%) is called “mild” and 0.2\pm0.1 (~95%) is called “significant”. It is preferable in the revised version to quote the credibility levels instead of (or in addition to) the subjective statements like “mild” or “significant”. Notice also that these estimates should also incorporate the theoretical uncertainty (grey band in Figure 1) of QUR. That said, the departure from 0 and departure from 0\pm 0.1 are different stories having different significances.

With respect to the previous comment, can the authors explain why there is a grey band for the \deltaf_2/f_2 and no such band for \delta\kappa/\kappa? According to the ideology presented in the paper, \delta\kappa/kappa is estimated from the PM data (eq.(2)) by inverting, so to say, QURs (see description in lines 110-118). If QUR f_2(kappa) has and intrinsic uncertainty then similar uncertainty is in the inverted relation \kappa(f_2). This should result in the vertical strip in fig.2 similar to grey band, and this uncertainty should also be taken into account in the estimation of the significance of the departure from null.

When the discrepancy between the numbers in the text and in the plot will be resolved, and significance levels recalculated taking into account the QUR uncertainties, the conclusions of the paper should be rechecked since they can potentially be altered.

The second point relates to Eq.(3).

The authors follow Ref.[26] and focus on the relative difference (i.e. \Delta f_2/f_2) instead of the absolute difference. In my opinion, this is redundant since there is no need to normalize the difference in order to discriminate between, say, f_2^{PM} and f_2^{IM}. But, of course, the authors can use relative difference if they wish.

What is more problematic, is that Eq.(3) (and similar Eq (6) of Ref.[26]) seems to lack exact probabilistic meaning. Integrand of Eq. (3) contains a product of two posteriors which are conditional of the same data (d_{IMPM}=d_IM+d_PM). Therefore, first of all, this is not a chain rule of conditional probabilities and these two terms are probability distributions conditional on the same data. The integrand can be correctly written as a product of two distributions if these two probability distributions (for (f_2, kappa_2) and for (\delta f_2,\delta \kappa_2)) are independent, which, from a first glance, would be a miracle coincidence.

Consider the imaginary situation when p(f_2, \kappa_2|d_{IMPM}) = p(\kappa_2|d_{IM})p(f_2|d_{PM}) (in reality there are some degree of correlation here). In that example the complete inspiral analysis does not add much and extracts f_2 from post-merger and \kappa_2 from pre-merger. However, Eq.(3) should work in this case as well. But in this case, the correct way to write is p(\epsilon_(f_2)|d_{IM},d_PM})=\int d f_2 d\kappa_2 p(\kappa_2|d_{IM})p(f_2|d_{PM}) \delta(epsilon – (f_2-QUR(kappa_2))/f_2), where QUR(kappa_2) is the tested relation. This does not reduce to eq. (3). Moreover, looking at eq.(3) one finds that it in some way contains p(f_2|d_{IM}) twice, i.e. it makes double use of the same data.

These arguments suggest that eq.(3) (and eq. (6) in Ref.[26]) are incorrect. I see two options here. Either the authors disprove my point (it is very probable that I am missing something) and prove that the probability distributions in the integrand of eq.(3) are independent, so the joint probability can be written as a product. Or, alternatively, they can stop at eq.(2) which does not suffer from these problems. Using of the absolute difference would not alter neither the test (after point one resolved) neither the conclusions. 

Apart from the concerns raised, there are a few minor points listed below (mostly, misprints).

 Minor points:

1. Line 31. Exponent and units are missing at \rho_sat.

2. Eqs. (2)-(3). Please define d_IM, d_PM, d_IMP.

3. Line 112. Eq.13 ->Eq.(13).

4. Line 122. posterior -> joint posterior; are -> is.

5. Eq.(3). On what data the posterior on \varepsilons is conditional? Cf. Eq. (2).

6. Figure 1. Caption. The panels are Left and Right, not Top and Bottom. Also “The binary are located”. Either “is located” or something plural should be added after “The binary”.

7. Line 149. Comma after while seems redundant. 

8. Line 150. affect -> affects.

9. Line 153. 3-sigma -> 3sigma.

10. Figure 2. Caption. Eq.1 -> Eq.(1); “Red lines” should be “green lines” in one instance. “identify”->”identifies”. “grey band report”->”grey band reports”.

11. Figure 2. Caption. What are the credibility contours here? Probably, the highest posterior density ones. Please indicate. The same for the grey band. 

12. Line 171. confidence -> credibility.

13. Line 176. extraced->extracted.

14. Line 179. invalidating? which invalidates?

15. Line 182. demonstrates->demonstrate. 

16. Line 186. Why is “phase transition” enclosed in the quotation marks? Isn’t the appearance of hyperons a phase transition. 

17. Line 199. Cf. -> cf.

18. Lines 197-199. I do not understand how this statement follows from the previous discussion. Can the authors expand the arguments a bit? What does “fine-tuning” mean in this context?

19. Some references to Phys Rev D are given as Phys Rev D year, volume and some as Phys Rev year, Dvolume. Please unify.

20. Reference 27 seemingly is published in Class.Quant.Grav., according to the arxiv preprint description (doi:10.1088/1361-6382/ab56290). Please update the reference.

Author Response

We thank the referee for their detailed report. We modified the draft according to the referee suggestions and include our comments to the referee report in blue. We resubmit the updated manuscript for publication in Particles.

Matteo Breschi

Reviewer 2 Report

This is a well written and interesting paper, that demonstrates how pre and post-merger signals can be used to test universal relations, and detect possible deviations in the EoS that may indicate softening due to exotica.

I recommend the paper for publication, and only have a minor typo to correct:

Fig.1 - the captions describes top and bottom panels when it should be left and right (it is confusing as the right panel does have top and bottom)

Author Response

We thank the referee for pointing out this typo. We resubmit the updated manuscript for publication in Particles.

On behalf of the authors, Matteo Breschi

Round 2

Reviewer 1 Report

I present my report in the pdf file attached.

Author Response

We thank the referee for the additional comments. We are glad that most of his concerns were satisfactorily addressed. We regret that our first response was not sufficiently detailed and clear to address the only remaining concern raised, and we now provide more detailed motivation for our approach, fully addressing this last concern.

The argument of the referee is based on the concern that (focusing on f only for simplicity) f_IM, f_PM and f_IMPM are not independent quantities. A counter-example to the general validity of Eq.(3) was provided.

First, it is correct that Eq.(3) is not valid for all conceivable class of models, including the extreme case suggested by the referee. This is because, if the two portions of data do not provide independent information, then the indepdence underlying Eq.(3) would be spoiled. However, Eq.(3) *is* valid on the set of models for which the IM and PM portions of data contain *independent* information, as in our case.

More specifically, the point here is that the three variables f_IM, f_PM and f_IMPM contain different physical information, since they are constructed from different models. In the IM case, the frequency contains information about the tidal polarizability of the stars. In the PM case, the frequency contains information about the postmerger dynamics. In the IMPM case instead, the frequency contains information about the tidal polarizability, the post-merger dynamics, quasi-universal relationships *and* the smooth transition between the two regimes, which constrains non-trivially the quantities involved. For these reasons, the quantities *are* independent, since their information content is. This implies that using Eq.(3) is justified.

The fact that part of the information is overlapping and that the different variables will contain correlations, as correctly mentioned by the referee, does not invalidate our reasoning. However, the  delta-correlation considered is simply unphysical and can never be realised independently of the SNR of the signal or the model considered. It is thus an unrealistic case that cannot be used to invalidate our analysis.

Finally, we stress that these same arguments underlie the statistical consistency test performed by the LIGO-Virgo-Kagra collaboration and used in O(10) peer-reviewed publications since 2015, including papers which are now milestones of the field, such as arxiv.org/abs/1602.03841. Our formalism is a different application of the same identical statistical approach, where the role of the polarizability etc. is replaced by the general relativistic mechanism underlying the emission of energy and angular momentum.

Round 3

Reviewer 1 Report

Third referee report on “Pre/post-merger consistency test for gravitational signals from binary neutron star mergers “ by Breschi et al.

Apparently, the authors did not take my concerns about Eq.(3) seriously. There are a lot of examples in science where a flaw in some equation was not revealed for quite a time. Citation count of the paper with incorrect equation does not help to validate it.

Of course, the delta-correlation is not realistic, however it was used by the referee as a simple illustrative model example. The real situation is closer to the delta-correlation pattern since one assumes that the involved methods work and “the best estimate” can not be completely independent of the “not the best estimate”, based on the one or another half of the data.

Notice that Eq.(3) requires for its validity the statistical independence, which is just the mathematical concept that the joint probability of two random variables is a product of the individual probabilities. This is not the methodological or philosophical question, it is a mathematical one. In this respect the phrase in response “The fact that part of the information is overlapping and that the different variables will contain correlations, as correctly mentioned by the referee, does not invalidate our reasoning.” means just the opposite. The overlapping information suggests absence of independence – the variables are linked by this joint portion of information. Until proved otherwise (by this I mean proved mathematically, not by vague general arguments), one can not assume independence and use the product form of the joint probability.

I wish to stress again, that, in my opinion, the artificial normalization is not needed for the test, which actually compare the difference with zero. The concept and results of the paper would not be harmed if this problematic normalization would not be taken.

With all due respect, I can not recommend the publication of the paper which contain a wrong (in my opinion, of course, but until disproved) equation, even if the same wrong equation appeared in some Ghosh et al. papers earlier, milestones them be or not!